# Conceptualising emotional and cognitive dysregulation amongst sports bettors; an exploratory study of 'tilting' in a new context

**Jamie Torrance**[ID]◉*, **Gareth Roderique-Davies**[ID]◉, **James Greville**[ID]◉,
**Marie O'Hanrahan**[ID]‡, **Nyle Davies**‡, **Klara Sabolova**‡, **Bev John**◉

Addictions Research Group, School of Psychology and Therapeutic Studies, University of South Wales, Pontypridd, United Kingdom

◉ These authors contributed equally to this work.
‡ MO, ND and KS also contributed equally to this work.
* jamie.torrance@southwales.ac.uk

## Abstract

Tilting is a poker-related phenomenon that involves cognitive and emotional dysregulation in response to unfavourable gambling outcomes. Tilting is characterised by an increase in irrational, impulsive and strategically weak betting decisions. This study aimed to adapt and investigate the concept of tilting amongst sport bettors in order to provide preliminary insight regarding previously unexplored instances of maladaptive sports betting. The sample consisted of 225 sports bettors who completed an online questionnaire that investigated their reported tilting episodes, awareness of tilting, impulsivity, perceived skill, gambling severity, gambling frequency, and product preferences. Cluster analyses revealed three distinct groups of sports bettors based on their reported tilting episodes and their awareness of this phenomenon. The first group were labelled 'Conscious tilters' due to being cognizant of their own tilting occurrence which was significantly higher than the other two groups. These 'Conscious tilters' had the highest mean problem gambling severity that was indicative of the 'problem gambler' categorisation. The second group were labelled 'Unconscious tilters' due to their underestimation of their own tilting occurrence and were categorised as 'moderate risk gamblers'. The third group were labelled 'Non-tilters' due to a relatively accurate perception of their low to non-existent tilting occurrence and were categorised as 'low-risk gamblers'. Additionally, there were significant differences between these groups in relation to reported gambling frequency, impulsivity, and product preferences. There is evidence of various classifications of 'tilters' within sports betting. Specific sports betting product features may also facilitate tilting and therefore require further research in this context. It is important for this research area to develop in order to mitigate harms associated with the rapidly changing sport betting environment.

**Data Availability Statement:** All relevant data are within the paper and its Supporting Information files (S1 Dataset).

**Funding:** This work was conducted as part of JT's PhD studies. The PhD is funded by GambleAware. The funders had no role in study design, data collection and analysis, decision to publish, or preparation of the manuscript. All other co-authors received no funding for their role in this research. https://www.begambleaware.org/.

**Competing interests:** The authors have declared that no competing interests exist.

# Introduction

Sports betting is a widely used mode of gambling that has seen a global increase in availability and complexity in recent years [1–4]. Due to the introduction of products such as in-play betting, and the instantaneous access provided by online bookmakers, there is evidence to suggest that sports betting is shifting into a more continuous and impulse-driven form of gambling [5–7]. Correspondingly, sports bettors may be particularly vulnerable to gambling-related harm due to their distinct characteristics and specific cognitions compared to non-sports gamblers. These traits primarily include the tendency to emotionally invest in betting (due to team loyalty), acting impulsively, and misperceiving gambling-related risk [8–11]. Although the identification of these characteristics within the literature is insightful, they are typically presented as separate constructs with little emphasis placed upon how they operate in combination.

An adaptable example of combining such constructs involves the poker-related phenomenon known as 'tilting'. This is defined as a state of frustration and irrationality when gambling due to experiencing repeated losses or being overwhelmed by strong negative emotions [12]. This state is characterised by a reduction in strategic or calculated gambling and an increase in aggressive and reckless bets [12–14]. The concept of tilting therefore encapsulates numerous cognitive-behavioural elements that are associated with gambling harm such as impulsivity, loss chasing, loss of control, emotional dysregulation, and irrational motives [11, 15, 16].

The current paper theorises that the concept of tilting can be appropriately mapped onto sports betting as a potential pathway towards gambling harm. Within this conceptual pathway, bettors may initially adopt a more rational and calculated strategy by placing low-risk bets at a low frequency. Following consecutive losses, poor referee decisions or depleting funds, sports bettors may begin to tilt in which they place (riskier) higher odds bets and/or spontaneously increase their betting frequency [17, 18]. Such decisions may be grounded in emotionality, impulsivity, and irrationality; harmful factors that have been outlined within previous studies of sports betting [19]. Maladaptive gambling behaviours and cognitions are often contextual given the vast structural differences across gambling modes and the demographic variations between the associated users [20, 21]. For example, investigating perceived skill in relation to scratch-card gamblers may prove problematic as scratch-cards are primarily considered to be governed by 'luck' and 'randomness' [22, 23]. Therefore, in attempting to adapt a poker-related concept into a sports betting context, it is imperative to first consider the fundamental similarities between these two gambling modes.

Sports betting and poker are both typically considered by bettors to involve a certain amount of skill that is grounded in contextual knowledge and gambling experience [16, 24, 25]. This may be due (in-part) to the incorporation of real-world information that engenders a sense of self-agency within these gambling modes. Poker players typically learn to 'read' the environment and the behaviours of their opponents whilst making economically underpinned decisions [26]. Similarly, sports bettors often utilise team/player/match statistics whilst evaluating odds-related information when placing bets [8]. Corresponding research evidence indicates that skill, knowledge, and expertise within both poker and sports betting may increase the frequency of successful bets [24, 26]. However, whether or not this initial advantage leads to an increase in tangible winnings is less clear due to the cyclic nature of the game mechanics that may encourage the re-staking of winnings [24, 26, 27]. Previous research has also highlighted the overlap between the traits and functional motives of sports bettors and players of 'skilled games' such as poker [28]. Primarily, both gambling modes are predominantly comprised of males who possess competitive traits [28].

As previously observed within poker [13, 29], tilting within sports betting may operate as one of many possible transitional factors that reshapes low-risk gambling into more harmful gambling behaviours. However, gamblers can only tilt when the structural characteristics of the gambling mode provide the opportunity to do so. To this end, the product features of in-play sports betting may be particularly facilitative of the cognitive-behavioural elements of tilting. These in-play product features include the ability to place numerous concurrent bets, instantly deposit more funds or the opportunity to bet on 'micro-events' that may have high odds-ratios [5, 30]. Consequently, an empirical understanding of which product features are most frequently used and deemed most important by in-play bettors is warranted by researchers when investigating tilting in this context. Such insight is important in relation to improving the regulation of particularly harmful gambling products; an issue that often transcends the personal control of the bettor.

Tilting can be measured indirectly by grouping together the primary indicators of tilting episodes via self-report scales. These typically involve issues associated with poor cognitive and emotional regulation within a gambling context [15]. However, it is likely that the associated harms are best mitigated when bettors possess the ability to identify and perceive tilting episodes [31, 32]. Therefore, it is useful to also measure the bettors' conscious perception of this phenomenon via direct questions. A similar approach was conducted in a previous study by Moreau, Sévigny, et al [15]. This study assessed tilting awareness amongst poker players by comparing scores from a self-report scale that indirectly assessed tilting episodes against direct questions that overtly measured the participant's awareness of this phenomenon. Three groups of participants were identified who all differed signifficantly in their awareness of tilting. These groups included poker players who overestimated their tilting, players who underestimated their tilting, and players who accuratlly reported little to no tilting [15]. The current study will explore tilting and the perceptions of sports bettors towards this phenomenon in line with the methodology utilized by Moreau, Sévigny, et al [15]. Adapting concepts from other forms of gambling into sports betting has been suggested to be particularly useful in highlighting transferable insight surrounding gambling-related harm [7]. Therefore, it is envisaged that adapting the poker-related concept of tilting within a sports betting context will provide an important contribution to the literature. Classifying sports bettors who are prone to tilting, investigating their conscious perception of this phenomenon, and comparing their characteristics will provide a preliminary understanding of the necessity of suitable harm-reduction strategies. To the best of our knowledge, the current study is the first to explore this topic within the sports betting sphere. Specifically, this study aimed to investigate:

1. How many classifications of sports bettors exist in relation to their reported tilting and awareness of this phenomenon?

2. How do these classifications of sports bettors differ in relation to gambling severity, gambling frequency, impulsivity, type of sport bettor (in-play/conventional), and perceived gambling skill?

3. What are the product preferences of the in-play bettors between the classifications?

## Methods

### Participants

The eligibility criteria of the current study required participants to be 18 years of age or older, reside in the UK, and have engaged with sports betting at least once in the previous 6 months. The decision to implement these criteria was underpinned by the need to investigate the

unique sports betting environment of the UK using this relatively short timeframe to minimise recall bias. The sample contained 225 sports bettors from England ($n = 154$), Wales ($n = 26$), Scotland ($n = 25$) and Northern Ireland ($n = 20$). Males constituted 79.11% ($n = 178$) of the sample with the largest proportion of participants (22.22%) representing the 18–24 age bracket ($n = 50$). In relation to educational level, the largest proportion of the sample (42.22%) had attained an undergraduate degree ($n = 95$). Ethnically, the largest proportion of the sample were represented by white individuals ($n = 184$, 81.78%). Asian ($n = 17$, 7.56%), Black ($n = 10$, 4.44%) and those of mixed ethnicity ($n = 14$, 6.22%) represented the remainder of the sample. Further demographic information is displayed in Table 1. The self-reported gambling behaviours of the sample indicate that 73.33% of participants engage with in-play betting ($n = 165$). In relation to gambling frequency, 44.44% gambled monthly ($n = 100$), 24% gambled weekly ($n = 54$), 22.67% gambled a few times a week ($n = 51$), and 8.89% gambled everyday ($n = 20$). The largest proportions of the sample participated in online sports betting ($n = 213$, 94.67%)

**Table 1. Demographic characteristics of the sample.**

| Demographic category | (n = 225) (%) |
|---|---:|
| **Gender** | |
| Male | 178 (79.11) |
| Female | 47 (20.89) |
| **Age range** | |
| 18–24 | 50 (22.22) |
| 25–29 | 43 (19.11) |
| 30–34 | 46 (20.44) |
| 35–39 | 43 (19.11) |
| 40–44 | 13 (5.78) |
| 45–49 | 13 (5.78) |
| 50–54 | 7 (3.11) |
| 55–59 | 7 (3.11) |
| 60–64 | 2 (0.89) |
| 64–69 | 0 |
| 70+ | 1 (0.44) |
| **Residing UK region** | |
| England | 154 (68.44) |
| Wales | 26 (11.56) |
| Scotland | 25 (11.11) |
| Northern Ireland | 20 (8.89) |
| **Ethnicity** | |
| White | 184 (81.78) |
| Asian | 17 (7.56) |
| Black | 10 (4.44) |
| Mixed ethnicity | 14 (6.22) |
| **Educational level** | |
| Primary school | 1 (0.44) |
| Secondary school | 26 (11.56) |
| College | 59 (26.22) |
| Undergraduate degree | 95 (42.22) |
| Postgraduate degree | 39 (17.33) |
| Other | 5 (2.22) |

and venue/bookmaker sports betting ($n$ = 91, 40.44%). Additional information relating to the gambling behaviour of the sample is displayed in Table 2.

## Measures

The study was conducted via an online self-report survey that initially required respondents to disclose some brief demographic information including age, gender, ethnicity, educational level, and residing country. Respondents were also asked to confirm their gambling frequency and what forms of gambling they engage with in both the online environment and at physical venues/bookmakers. As a subjective measure of perceived tilting frequency, the next section within the survey included a description of tilting and the following question: "*According to you, how many times have you experienced 'tilting' whilst betting on sports in the last 6 months*?". This question was measured via a 7-point Likert scale ranging from "0 times" to "more than 10 times". Respondents were then asked whether their sports betting involved in-play betting as a binary measure to separate in-play and conventional sports bettors. The subsequent sections of the survey incorporated the following measures.

**In-play betting product feature scale.** Only respondents who confirmed that their sports betting involved in-play betting completed this scale within the survey. Conventional sports bettors were diverted to the next section. The in-play betting product feature scale was developed and underpinned by a brief scoping exercise that involved a literature search and

**Table 2. Gambling behaviours of the sample.**

| Gambling characteristic | (n = 225) (%) |
|---|---|
| **Type of sports bettor** | |
| In-play | 165 (73.33) |
| Conventional | 60 (26.67) |
| **Gambling frequency** | |
| Everyday | 20 (8.89) |
| A few times a week | 51 (22.67) |
| Weekly | 54 (24.00) |
| Monthly | 100 (44.44) |
| **Participation in online gambling[a]** | |
| Sports betting | 213 (94.67) |
| Casino & table games (blackjack, poker etc.) | 70 (31.11) |
| Scratch cards | 39 (17.33) |
| Lottery | 79 (35.11) |
| Bingo | 16 (7.11) |
| Gaming/slot machines | 51 (22.67) |
| Other | 8 (3.56) |
| **Participation in venue/bookmaker gambling[a]** | |
| Sports betting | 91 (40.44) |
| Casino & table games (blackjack, poker etc.) | 25 (11.11) |
| Scratch cards | 23 (10.22) |
| Lottery | 29 (12.89) |
| Bingo | 11 (4.89) |
| Gaming/slot machines | 21 (9.33) |
| Other | 5 (2.22) |

[a]Respondents could choose more than one answer.

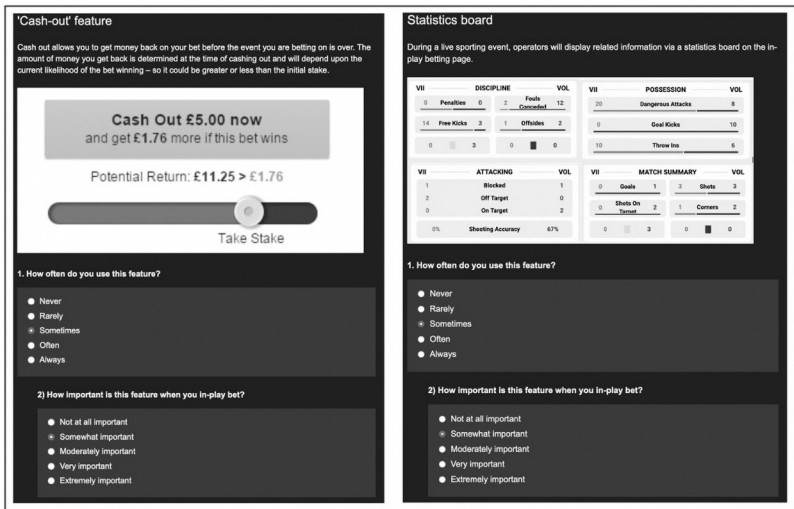

**Fig 1. Screenshot examples of the in-play betting product feature scale.**

observations of in-play betting pages provided by some of the most popular UK-based gambling operators (Paddy Power, William Hill, Bet365, Betfair, Betfred and Ladbrokes). This process was performed by JT, MOH, and involved consultations with the wider research team to reach a consensus concerning the pertinence of the features included within the scale. Consequently, the following product features were deemed to be the archetypal and distinctive elements that in-play bettors can engage with; 1) embedded live stream, 2) virtual live updates, 3) a statistics board, 4) the 'cash-out' feature, 5) instant depositing of funds, 6) the ability to place concurrent bets, 7) the ability to place high-odds bets. As indicated within Fig 1, each product feature was presented alongside a brief description, a picture, and two preference-related questions measured via 5-point Likert scales; "*How often do you use this feature?*" (never-always) and "*How important is this feature when you in-play bet?*" (not at all important-extremely important).

**Short (SUPPS-P) Impulsivity Scale.** The current study utilised the shortened version of the UPPS-P Impulsivity Scale [33, 34]. This 20-item self-report questionnaire assesses impulsivity across five subscales that address negative urgency, positive urgency, lack of perseverance, lack of premeditation and sensation seeking. Items are measured in relation to agreement on a 4-point Likert scale ranging from 1 (strongly disagree) to 4 (strongly agree). Scores across the subscales can be summed (after reverse scoring 12 items) and are indicative of higher levels of trait impulsivity. The shortened scale is considered a valid and reliable alternative to the longer (57-item) scale with comparable inter-correlations and overlapping variance between the two versions [33]. Within the current study, the Cronbach's alpha coefficient of internal consistency for the SUPPS-P was considered acceptable ($\alpha = .70$) to good ($\alpha = .80$) across the 4-item subscales.

**Problem Gambling Severity Index (PGSI).** The PGSI is a widely used instrument that adopts a 9-item self-report scale that measures problem gambling severity [35]. Total scores can be produced that correspond to four clinical categories that include non-problem, low risk, moderate risk, and problem gambler. In relation to the psychometric properties of the PGSI, the scale possesses good internal consistency, test-retest reliability, and factor structure [35, 36]. Although the literature concerning the validity of the PGSI is less clear, it is deemed a useful and applicable tool that is psychometrically stronger than other similar scales [37].

Within the current study, the Cronbach's alpha coefficient of internal consistency for the PGSI was considered excellent ($\alpha$ = .93).

**9-item Online Poker Tilting Scale (OPTS-9).** The 9-item Online Poker Tilting Scale (OPTS-9) is a shortened version of the original 17-item scale that was validated within a French population [38, 39]. The OPTS-9 was adapted within the current study to provide an additional measure of tilting occurrence without explicitly discussing this phenomenon. The items are measured on a five-point Likert scale (never, rarely, sometimes, often, almost every time) with higher summed scores indicative of higher tilting occurrence. The key aspects of tilting that are covered by the items within the OPTS-9 include risk-taking, alteration of focus, desire to win, dissociation, loss of control, frustration, negative mood, irritability/anger, and aggressive actions. Within the scale, these items are grouped under two labels; either emotional or cognitive dysregulation. Due to this poker-related scale being adapted to sports betting in this context, each question was prefaced with '*while sports betting. . .*' rather than '*while playing poker. . .*'. Additionally, the examples described within the frustration item ("*I feel frustrated due to bad luck, other players' behaviour etc*") were changed to "*I feel frustrated due to bad luck, poor referee decisions / team performance etc)*". All other items within the scale remained unaltered as they were not poker specific in their wording. Within the current study, the Cronbach's alpha coefficient of internal consistency for the OPTS-9 was considered excellent ($\alpha$ = .90).

**GamCog Perceived Gambling Skill (PGS) subscale.** Self-reported perceptions of gambling-related skill were measured using the Perceived Gambling Skill (PGS) subscale of the GamCog [40]. The GamCog is an instrument used in the assessment of cognitions within a gambling or video gaming context. The PGS subscale is comprised of 5 items that are scored on a 7-point Likert scale ranging from 1 (strongly disagree) to 7 (strongly agree). The PGS was deemed most fitting in measuring perceived skill within the current study in comparison to other similar scales such as the Gambling-related Cognitions Scale (GRCS) [41]. This decision was underpinned by the way in which the PGS specifically focuses upon perceived gambling-related skill in a manner that aligns appropriately with sports betting. In contrast, the GRCS focuses upon predictive control and illusions of control rather than perceived skill which may lead to overestimations of cognitive distortion in skills-based gambling modes such as poker or sports betting [42]. Within the current study, the Cronbach's alpha coefficient of internal consistency for the PGS was considered good ($\alpha$ = .88).

## Procedure

Data collection occurred between May and July 2021 and was conducted via a survey that was hosted online. Invitations to participate were posted on Twitter, Reddit and via email. In relation to Reddit, the survey was most widely distributed via the "Soccer", "Casual UK", and "Problem Gambling" forums (subreddits). All of the respondents gave informed written consent to participate and gave permission for their anonymised data to be utilised within study reports, peer-reviewed publications and displayed via open science data sharing platforms. Participation was voluntary and respondents were not financially incentivised. Due to this lack of incentivisation alongside sports bettors often displaying impulsive traits [8, 10], short versions of the scales were incorporated to reduce attrition rates and time expenditure when completing the survey. The structure of the survey involved the following sections: 1) demographic information and gambling behaviour; 2) perceived tilting frequency (past 6 months); 3) separating in-play and conventional sports bettors 4) in-play betting product feature scale; 5) SUPPS-P Impulsivity Scale; 6) PGSI; 7) OPTS-9; 8) GamCog PGS. Ethical Approval for this study was given by the School of Psychology and Therapeutic Studies ethics panel at The

University of South Wales. All procedures were in accordance with the standards of The University of South Wales and with the 1964 Helsinki Declaration and its later amendments or comparable ethical standards.

## Data analysis

The following analytical procedures were conducted using IBM SPSS Statistics version 27. To create distinct classifications of participants based on their reported tilting episodes and awareness of this phenomenon, a cluster analysis was performed. The process of clustering allows for the formulation of groups that are higher in homogeneity within themselves whilst being heterogenous between each other [43]. Firstly, z-scores ($M = 0$, $SD = 1$) were calculated in order to standardise the two tilting variables (total OPTS-9 scores and total perceived tilting) for the sake of comparability [44]. Subsequently, these z-scores were subjected to hierarchical clustering to help define the number of suitable clusters within the sample before further analysis. Hierarchical clustering involves the creation of a nested sequence of partitions that includes an all-encompassing single cluster at the top of the hierarchy with clusters comprised of single cases at the bottom. Each level within the hierarchy is comprised of the two lower clusters beneath it and can be displayed graphically via a hierarchical tree known as a dendrogram. Ward's method of agglomerative clustering was utilised throughout this initial process using the squared Euclidean distance. Following this, the solution of the hierarchical clustering was determined and incorporated into a k-means cluster analysis. The k-means algorithm involved the previously calculated z-scores of the perceived tilting and OPTS-9 total score variables in order to allocate each participant into a cluster.

The stability and sufficiency of the k-means cluster model was validated via a series of ANOVAs and post hoc tests (Tukey and Games-Howell) that investigated differences between the clusters in terms of the two tilting variables (unstandardised OPTS-9 and perceived tilting scores). These ANOVAs and post hoc tests were also performed to investigate the differences between clusters with regards to characteristics such as gambling severity (PGSI), impulsivity (SUPPS-P), and perceived skill (PGS). To investigate the association between cluster membership and the type of sports betting the participants engage with (in-play or conventional), Chi-square analysis was conducted. Differences in gambling frequency and product preferences between the clusters were analysed using the non-parametric Kruskal-Wallis (one-way ANOVA on ranks) test with Dunn-Bonferroni post-hoc testing.

## Results

### Tilting cluster formulation and characteristics

The hierarchical cluster analysis allowed for the identification of 3 distinct clusters amongst the sample. This classification was formulated using the agglomeration schedule alongside observations of the dendrogram. The standardised z-scores for each tilting variable (OPTS-9 and perceived tilting) across the 3 clusters are displayed in Fig 2 as a result of K-means clustering. The participants in Cluster 1, labelled 'Conscious tilters' ($n = 24$) had an OPTS-9 z-score of 1.47 and a total perceived tilting z-score of 2.36. Cluster 2 labelled 'Unconscious tilters' (n = 71), contained participants with an OPTS-9 z-score of .72 and a total perceived tilting z-score of .09. The participants in Cluster 3, labelled 'Non-tilters' ($n = 130$) had an OPTS-9 z-score of -.67 and a total perceived tilting z-score of -.49.

Between the clusters, the mean impulsivity (SUPPS-P) and perceived skill (PGS) scores met the assumption of homogeneity of variance via Levene's $F$ testing and were subjected to one-way ANOVAs. Welch tests were performed to analyse the differences between the mean OPTS-9, perceived tilting and PGSI scores as these failed to meet the assumption of

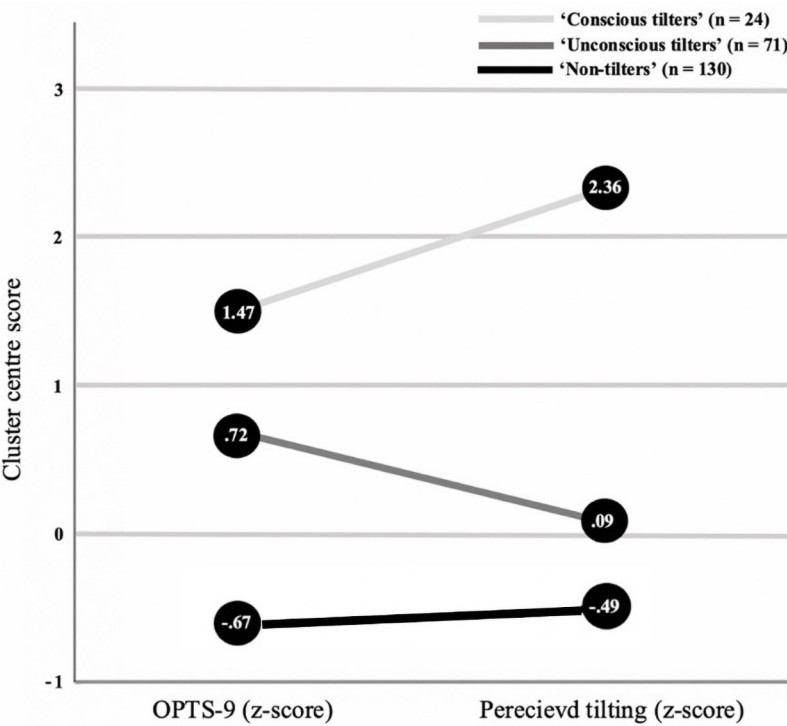

**Fig 2. Cluster centre z-scores of OPTS-9 and perceived tilting.**

homogeneity of variance. Effect sizes were calculated using an estimation of omega squared. The results of these analyses are presented within Table 3 in addition to the unstandardised means and standard deviations of the measured variables. The Welch tests and ANOVAs revealed statistically significant main effects ($p < .001$) that indicated the 3 clusters differed in relation to mean OPTS-9 (*est.* $\omega^2$ = .60), perceived tilting (*est.* $\omega^2$ = .54), PGSI (*est.* $\omega^2$ = .25) and SUPPS-P scores (*est.* $\omega^2$ = .14). The mean PGS scores were not significantly different between the clusters.

Post-hoc analysis involved Tukey tests for the variables that met the assumption of homogeneity of variance. Games-Howell testing was utilised for variables that violated this assumption. As denoted next to the mean scores displayed in Table 3, Cluster 1 scored significantly

**Table 3. Comparisons of the characteristics between the three clusters.**

| | Cluster 1 'Conscious tilters' (n = 24) | Cluster 2 'Unconscious tilters' (n = 71) | Cluster 3 'Non-tilters' (n = 130) | dif 1 | dif 2 | F | est. $\omega^2$ |
|---|---|---|---|---|---|---|---|
| **Psychometric characteristic** | M (SD) | M (SD) | M (SD) | | | | |
| **OPTS-9** | 15.79 (7.04) [2,3] | 11.32 (3.85) [1,3] | 2.99 (2.19) [1,2] | 2 | 50.55 | 170.74* | .60 |
| **Perceived tilting** | 3.83 (1.13) [2,3] | 1 (.68) [1,3] | .28 (.50) [1,2] | 2 | 53.04 | 134.89* | .54 |
| **PGSI** | 9 (6.95) [2,3] | 4.76 (4.88) [1,3] | 1.02 (1.55) [1,2] | 2 | 47.95 | 34.19* | .25 |
| **SUPPS-P** | 47.96 (7.50) [2,3] | 43.45 (6.51) [1,3] | 39.61 (6.56) [1,2] | 2 | 222 | 19.53* | .14 |
| **PGS** | 20.04 (7.20) | 19.90 (6.00) | 17.76 (6.86) | 2 | 222 | 2.60 | |

* $p < .001$.
[1, 2, 3] Representative of cluster that is significantly different ($p < .05$).

higher across 4 of the 5 psychometric characteristics compared to Cluster 2; OPTS-9 ($p$ = .017); Perceived tilting ($p$ < .001); PGSI ($p$ = .025); SUPPS-P ($p$ = .012). Cluster 1 was also significantly higher across the mean scores of these psychometric characteristics compared to Cluster 3; OPTS-9 ($p$ < .001); Perceived tilting ($p$ < .001); PGSI ($p$ < .001); SUPPS-P ($p$ < .001). Similarly, Cluster 2 also displayed significantly higher mean scores across these variables compared to Cluster 3; OPTS-9 ($p$ < .001); Perceived tilting ($p$ < .001); PGSI ($p$ < .001) and SUPPS-P ($p$ < .001).

A chi-square test of independence was performed to examine the relation between cluster membership and the type of sports betting the participants engaged with. Participants were categorised as either in-play bettors (n = 165) or conventional sports bettors (n = 60). This analysis revealed that there was no significant association between cluster membership and sports betting type, $\chi^2(2)$ = 1.44, $p$ = .49. In order to analyse any differences in gambling frequency (monthly, weekly, a few times a week, everyday) between the clusters, a Kruskal-Wallis H-test was performed. The results of this test indicated that there was a statistically significant difference in gambling frequency between at least one pair of clusters, $H$ = 13.23, $p$ = .001. The corresponding effect size was calculated via the eta squared measure using the Kruskal-Wallis H statistic, $\eta^2_H$ = .05 [45]. A mean rank gambling frequency of 154.56 was observed for Cluster 1, 113.48 for Cluster 2 and 105.07 for Cluster 3. Post-hoc analysis involved Dunn-Bonferroni testing that revealed the differences in gambling frequency were statistically significant between Cluster 3 and Cluster 1 ($p$ = .001), alongside Cluster 2 and Cluster 1 ($p$ = .014). However, differences in gambling frequency were not statistically significant between Cluster 3 and Cluster 2.

Overall, these results indicate that there were significant differences observed between the clusters in relation to OPTS-9 scores, perceived tilting, PGSI scores, SUPPS-P scores, and gambling frequency. Contrastingly, there were no significant differences observed in relation to PGS (perceived skill) scores and gambling type.

## In-play product preferences

To analyse the product preferences (frequency of use and perceived importance) of the in-play bettors ($n$ = 165) between the clusters, Kruskal-Wallis $H$-tests were performed. The results of these tests indicated that there were statistically significant differences in product preferences between at least one pair of clusters across 5 of the 14 product preferences. These included: the frequency at which the embedded livestream feature was used ($H$ = 6.52, $p$ = .038); the perceived importance of the virtual updates feature ($H$ = 11.38, $p$ = .003); the frequency at which the statistics board was used ($H$ = 11.69, $p$ = .003); the frequency at which the instant deposit feature was used ($H$ = 18.39, $p$ < .001); the perceived importance of the instant deposit feature ($H$ = 9.63, $p$ = .008). The results and effect sizes for these analyses are displayed in Table 4. The effect sizes were calculated via the eta squared measure ($\eta^2_H$) using the Kruskal-Wallis $H$ statistic [45].

Post-hoc analysis consisted of Dunn-Bonferroni testing. In relation to the embedded livestream feature, Cluster 1 (*mean rank* = 103.35) reported a significantly higher frequency of use compared to Cluster 3 (*mean rank* = 76.10), $p$ = .048. Concerning the statistics board feature, Cluster 1 (*mean rank* = 103.90) reported a significantly higher frequency of use compared to Cluster 3 (*mean rank* = 72.43), $p$ = .018. Similarly, Cluster 2 (*mean rank* = 93.87) also reported using the statistics board at a significantly higher frequency compared to Cluster 3 (*mean rank* = 72.43), $p$ = .023. With regards to the instant deposit feature, Cluster 1 (*mean rank* = 104.80) reported a significantly higher frequency of use compared to Cluster 3 (*mean rank* = 69.56), $p$ = .006. Similarly, Cluster 2 (*mean rank* = 98.65) also reported using the instant deposit feature

**Table 4. Kruskal-Wallis _H_ tests of product preferences between clusters.**

| | Cluster 1 _'Conscious tilters'_ (n = 20) | Cluster 2 _'Unconscious tilters'_ (n = 52) | Cluster 3 _'Non-tilters'_ (n = 93) | _n_ [a] | _dif_ | _H_ | $\eta^2_H$ |
|---|---|---|---|---|---|---|---|
| **Frequency of use** | Mean ranking | Mean ranking | Mean ranking | | | | |
| Embedded livestream | 103.35[3] | 87.51 | 76.10[1] | 165 | 2 | 6.52* | .03 |
| Virtual live updates | 94.70 | 91.10 | 75.96 | 165 | 2 | 5.01 | |
| Statistics board | 103.90[3] | 93.87[3] | 72.43[1,2] | 165 | 2 | 11.69* | .06 |
| Cash-out feature | 92.50 | 83.76 | 80.53 | 165 | 2 | 1.16 | |
| Instant deposit | 104.80[3] | 98.65[3] | 69.56[1,2] | 165 | 2 | 18.39** | .10 |
| Concurrent bets | 91.38 | 87.00 | 78.96 | 165 | 2 | 1.75 | |
| High-odds / microevents | 99.48 | 89.09 | 76.05 | 165 | 2 | 5.60 | |
| **Importance** | Mean ranking | Mean ranking | Mean ranking | | | | |
| Embedded livestream | 73.47 | 74.48 | 63.44 | 136 | 2 | 2.70 | |
| Virtual live updates | 93.09[3] | 79.38 | 61.82[1] | 142 | 2 | 11.38* | .07 |
| Statistics board | 91.18 | 81.74 | 69.73 | 152 | 2 | 5.12 | |
| Cash-out feature | 74.42 | 75.19 | 78.66 | 153 | 2 | .29 | |
| Instant deposit | 94.75[3] | 86.37[3] | 67.67[1,2] | 154 | 2 | 9.63* | .05 |
| Concurrent bets | 76.88 | 64.96 | 65.14 | 132 | 2 | 1.41 | |
| High-odds / microevents | 66.75 | 69.22 | 63.75 | 131 | 2 | .60 | |

[a]Participants who reported 'never' using a feature did not rate its importance.

[1, 2, 3]Representative of cluster that is significantly different ($p < .05$).

* p < .05,

**p < .001.

at a significantly higher frequency compared to Cluster 3 (_mean rank_ = 69.56), _p_ = .001. The perceived importance of the instant deposit feature reported by Cluster 1 (_mean rank_ = 94.75) was significantly higher than Cluster 3 (_mean rank_ = 67.67), _p_ = .036. The perceived importance of the instant deposit feature was also significantly higher amongst Cluster 2 (_mean rank_ = 86.37) compared to Cluster 3 (_mean rank_ = 67.67), _p_ = .044. Lastly, in relation to the virtual live updates feature, Cluster 1 (_mean rank_ = 93.09) reported this feature to be significantly higher in importance compared to Cluster 3 (_mean rank_ = 61.82), _p_ = .010. Overall, the preferences differed significantly across some but not all of the in-play product features between the three clusters.

## Discussion

The current study aimed to contribute to the literature by adapting and investigating the concept of 'tilting' amongst sports bettors within the gambling environment of the UK. The evidence here indicates that tilting is indeed observable in this context with distinct groups of sports bettors who differ according to their reported tilting occurrence and awareness of this phenomenon. Specifically, cluster analyses distinguished three groups who reported significantly different OPTS-9 scores and perceived tilting frequencies in congruence with Moreau, Sévigny, et al [15]. These groups also differed significantly in relation to gambling severity, gambling frequency, and impulsivity. Significant differences were not observed in relation to perceived skill or type of sports bettor (in-play or conventional). Lastly, the in-play bettors across these groups differed significantly in relation to some but not all of their reported product preferences.

The first group were labelled 'Conscious tilters' and consisted of 24 bettors who reported significantly higher OPTS-9 scores and perceived tilting frequencies compared to the two

other groups. These 'Conscious tilters' were labelled as such due to high z-scores in relation to the two tilting variables. Specifically, with a cluster centre z-score of 2.36 for perceived tilting frequency and 1.47 for OPTS-9 scores, it is evident that individuals within this group are highly cognizant of their own tilting episodes. This group also reported significantly higher mean PGSI scores compared to the other two groups. With a mean PGSI score of 9, this 'Conscious tilting' group are representative of the 'problem gambler' categorisation [35]. Consequently, this finding supports the notion that tilting may operate as a facilitator of harmful or maladaptive sports betting [12, 15]. Similarly, with significantly higher gambling frequencies (mean rank = 154.56) compared to the two other groups, these individuals may be more prone to tilting as more frequent engagements with the gambling environment may provide more opportunities to tilt. This particular group are likely situated at the higher end of the harm-spectrum and their tilting may be facilitated (in-part) by their significantly higher mean impulsivity scores (47.96) compared to the two other groups. Tilting is suggested to be grounded in uncalculated and reckless betting decisions that are often the product of irrational motives and acting in a spontaneous or overly emotional manner [12]. Therefore, it is unsurprising that those who are most prone to tilting also report significantly higher levels of impulsivity [46]. This finding was not previously observed amongst poker players [15] suggesting that the nature of maladaptive sports betting may be more impulse-driven and/or that sports bettors may possess more impulsive traits in comparison [19]. Overall, these 'Conscious tilters' represent a cohort of sports bettors who, despite being highly aware, appear to experience the highest tilting occurrence alongside higher levels of associated gambling-harm.

The second group were labelled 'Unconscious tilters' and consisted of 71 bettors who reported significantly higher OPTS-9 scores and perceived tilting frequencies compared to the third group. This particular group were labelled in accordance with the discrepancy between the two reported tilting variables. Specifically, these 'Unconscious tilters' reported a lower cluster centre z-score for perceived tilting frequency (.09) in comparison to a higher OPTS-9 z-score (.72). This discrepancy is indicative of a low conscious awareness or underestimation of tilting occurrence. In addition, these 'Unconscious tilters' reported a significantly higher mean PGSI score (4.76) compared to the third group and would therefore be categorised as 'moderate-risk gamblers' [35]. The same trend was also observable in relation to the mean impulsivity scores (43.45) in comparison to the third group. In combination, these factors are particularly important when considered via the lens of harm-reduction. Given that tilting may operate as a gateway to maladaptive or disordered gambling [47], it is necessary to recognise that this group are represented by a sizeable portion of the sample (31.56%) compared to the 'Conscious tilting' group (10.67%). In summary, the current study provides preliminary evidence for a cohort of sports bettors who are 'unconscious' or unaware of their own tilting episodes. At face value, this cohort appears to be distinct and detached from the other groups. However, it could be proposed that this group may operate as a steppingstone into the 'Conscious tilting' group for some bettors. This transition may take place if tilting episodes and the associated harms become too frequent and intense to misperceive. Further investigation into this potential transition as a future research priority is therefore recommended.

The third group were labelled 'Non-tilters' and consisted of 130 participants. This group were labelled according to their significantly lower OPTS-9 scores and perceived tilting frequencies compared to the two other groups. The cluster centre z-scores for the OPTS-9 (-.67) and perceived tilting frequency (-.49) within this particular group are indicative of a relatively accurate perception of little to no tilting episodes. Correspondingly, this group reported a significantly lower mean PGSI score (1.02) in comparison to the two other groups which is representative of the 'low-risk gambler' categorisation [35]. The mean impulsivity score (39.61) of this group was also significantly lower than the two other groups. Overall, these 'Non-tilters'

represent a cohort of sports bettors who report the lowest tilting occurrence, are less impulsive, gamble at the lowest frequency and experience the lowest amount of gambling-related harm.

In light of the behavioural and psychological characteristics that are encompassed by the concept of tilting, the current findings suggest that maladaptive sports betting can be investigated via the emergent paradigm of emotional and cognitive dysregulation. Although this approach has been recently adopted in the prediction of maladaptive gambling in general [48, 49], it appears particularly pertinent to sports betting. For example, sports betting may be more emotionally charged than other forms of gambling due to the sports fan identity and the emotional investment that bettors place upon their respective team to succeed [50]. In addition, the analytical nature of sports betting combined with a high number of betting opportunities may facilitate erroneous cognitions, impulsivity, and loss-chasing [11, 51]. However, tilting is currently a more established and recognised concept amongst poker players compared to sports bettors. Within the poker sphere, the frequency of tilting is typically associated with perceived skill as players who wish to increase their expertise are expected to display adequate emotional regulation when gambling [26, 52]. In contrast, the three groups of sports bettors within the current study did not differ significantly in relation to their perceived gambling skill when clustered according to their tilting scores and perceived tilting frequency. Although placing a new emphasis upon identifying tilting episodes and the importance of emotional regulation would likely benefit sports bettors to an extent, the structural characteristics and product features associated with this gambling mode are also very relevant in this context.

The evolution of the sports betting environment is rapid, and the associated products are increasing in complexity and availability [5, 7]. Therefore, the current study also investigated the product preferences of the in-play bettors across the three groups. The ability to instantly deposit funds was the only in-play product feature that differed significantly between the groups in terms of both frequency of use and perceived importance. The 'Conscious tilters' and 'Unconscious tilters' within the current study both reported using this in-play feature at a significantly higher frequency compared to the 'Non-tilters'. The instant depositing of funds has been previously highlighted as a catalyst in prolonging sports betting sessions alongside possessing the potential to facilitate impulsive gambling behaviours [5]. Theoretically, the instant deposit feature may enable tilting bettors to immediately replenish their lost funds in order to place more reckless and impulse-driven bets during episodes of irrationality.

The 'Conscious tilters' and 'Unconscious tilters' also reported using the statistics board feature at a significantly higher frequency compared to the 'Non-tilters'. Similarly, the 'Conscious tilters' reported using the embedded livestream feature at a significantly higher frequency and deemed the virtual live updates feature to be significantly more important compared to the 'Non-tilters'. The statistics board, embedded livestream, and live update features can all be categorised as information-based. These features provide bettors with either visual or statistical information related to the respective game, match, or event. It appears rational to assume that in-play bettors utilise these features to inform their sports betting decisions although whether they are used before, during, or after tilting episodes remains unclear. Previous research has suggested that information-based product features may facilitate illusions of control amongst sports bettors by providing them with a perceived advantage [53]. Supporting evidence for this notion is provided here given these features were most frequently used and deemed most important by those with higher problem gambling severity and tilting occurrence within the current study. However, there was no significant association between cluster membership and sports betting type (in-play or conventional). Therefore, the product preferences of the in-play bettors should be interpreted conservatively in relation to the facilitation of tilting in this context. In light of these preliminary findings, future research should aim to investigate the influence of sports betting product features in facilitating tilting episodes within more naturalistic settings.

This exploratory study has also raised numerous questions relating to the potential context in which tilting episodes take place. For example, sports betting and alcohol products are often marketed in tandem [54, 55] and it is common for sports bettors to consume alcohol whilst gambling [56]. Given the evidence indicating the detrimental effect of alcohol upon rational decision making and emotional regulation [57–59], future research should seek to investigate such contextual factors and their relationship with tilting episodes amongst sports bettors. It is likely that tilting is a multifaceted phenomenon that is facilitated by game design, cognition, and environmental factors [32]. Therefore, research that aims to investigate tilting via multiple avenues from a harm-reduction perspective is warranted.

### Limitations

There are some potential limitations that should be considered in light of the current findings. Firstly, this study incorporated questions that required participants to retrospectively identify the frequency of their own tilting episodes. Given that these questions typically rely upon the accurate recollection of the participants, there is potential for recall bias. However, the questions were framed to assess tilting over the relatively short time-frame of the previous 6-months in an attempt to minimise this bias. Furthermore, the concept of tilting was investigated via an additional measure (OPTS-9) that did not include a time-frame in order to provide a more holistic assessment of this phenomenon. A second limitation relates to the development of the in-play product feature scale. It is possible that certain in-play product or structural features were not included within this scale as they are not entirely consistent across operator websites and apps. To increase the rigor and applicability of this scale, the archetypical in-play product features were incorporated after a consensus had been reached by the research team concerning their inclusion.

### Conclusions

In contribution to the international gambling literature, the current findings provide preliminary evidence for the existence of tilting amongst sports bettors. The results indicate that there are at least three profiles of sports bettors who differ in relation to their tilting scores alongside their perception and awareness of this phenomenon. There are characteristic differences between these groups that include variations in gambling severity, gambling frequency, impulsivity, and product preferences. Specifically, there is evidence here to suggest that tilting may operate as a facilitator of maladaptive sports betting or increased gambling severity, although further research is warranted in relation to this interaction. The findings of this exploratory study also open up new lines of enquiry regarding the role of emotional and cognitive dysregulation within maladaptive sports betting; two elements that are encompassed by the concept of tilting. Consequently, there is a need to further investigate the influence of specific product features upon the onset and maintenance of tilting episodes. The preliminary findings here suggest that the ability to instantly deposit funds and utilise sports/odds-related information may facilitate tilting. It is important for this research area to develop in step with the rapid expansion and increasing complexity of the sports betting sphere. Gaining insight into tilting as a contextual factor that reshapes low risk into maladaptive sports betting is beneficial to gamblers, researchers, and service providers.

### Supporting information

**S1 Dataset. The raw quantitative data of the current study.** Values and labels are also included.
(XLSX)

## Author Contributions

**Conceptualization:** Jamie Torrance, Gareth Roderique-Davies, James Greville, Bev John.

**Data curation:** Jamie Torrance.

**Formal analysis:** Jamie Torrance.

**Investigation:** Jamie Torrance.

**Methodology:** Jamie Torrance, Gareth Roderique-Davies, James Greville, Marie O'Hanrahan, Nyle Davies, Klara Sabolova, Bev John.

**Software:** Jamie Torrance.

**Supervision:** Gareth Roderique-Davies, James Greville, Bev John.

**Writing – original draft:** Jamie Torrance.

**Writing – review & editing:** Jamie Torrance, Gareth Roderique-Davies, James Greville, Marie O'Hanrahan, Nyle Davies, Klara Sabolova, Bev John.

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
