## [Decision Letter · Decision Letter 0]

19 Jan 2022

PONE-D-21-32602Conceptualising emotional and cognitive dysregulation amongst sports bettors; an exploratory study of ‘tilting’ in a new contextPLOS ONE

Dear Dr. Torrance,

Thank you for submitting your manuscript to PLOS ONE. After careful consideration, we feel that it has merit but does not fully meet PLOS ONE’s publication criteria as it currently stands. Therefore, we invite you to submit a revised version of the manuscript that addresses the points raised during the review process.

We look forward to receiving your revised manuscript.

Kind regards,

Rabiu Muazu Musa, PhD

Academic Editor

PLOS ONE

Journal Requirements:

Reviewers' comments:

Reviewer's Responses to Questions

**Comments to the Author**

1. Is the manuscript technically sound, and do the data support the conclusions?

Reviewer #1: Yes

2. Has the statistical analysis been performed appropriately and rigorously? 

Reviewer #1: I Don't Know

3. Have the authors made all data underlying the findings in their manuscript fully available?

Reviewer #1: Yes

4. Is the manuscript presented in an intelligible fashion and written in standard English?

Reviewer #1: Yes

5. Review Comments to the Author

Reviewer #1: This paper deals with an interesting topic and is generally a pleasure to read. Please see below for my comments and I hope they can help the authors improve or refine their manuscript.

* On page 5 (line 103): Please correct “Titling”.

* On page 6 (line 139): Please double check 20.44 there and 20.22 in table 1, and ensure consistent and correct information be reported in these places.

* On page 6 (line 146): Please double check 20.67 there – would this need to be revised to 51 based on table 2?

* On page 8 (line 208, 210-211): “in-play betting product feature scale” was mentioned there as well as in line 290 on page 11, but Fig. 1 title mentions “in-play product preferences scale” – please try to use consistent wording throughout the paper.

* On page 9 (line 248, 249): “Nine Item Online Poker Tilt Scale” and “The 9-item Online Poker Tilting Scale” were mentioned there – please try to use consistent wording throughout the paper.

* On page 10 (line 254): “including” might need revision.

* On page 14 (line 366): one of the two “that” needs to be deleted.

* On page 16 (line 409): first “between” needs to be revised to better wording.

* On page 20 (line 517-518): The sentence “However … groups.” needs to be revised.

* Some of the references in the reference list might have incomplete or incorrect information, or parts that need revision (e.g., reference 4, 47). Please double check all references and make sure their information is correct.

6. PLOS authors have the option to publish the peer review history of their article (what does this mean?). If published, this will include your full peer review and any attached files.

Reviewer #1: No

---

## [Author Response · Author response to Decision Letter 0]

21 Jan 2022

Dear Editor, 

Thank you very much for the review of our manuscript entitled: “Conceptualising emotional and cognitive dysregulation amongst sports bettors; an exploratory study of ‘tilting’ in a new context”. We sincerely appreciate all valuable comments and suggestions from you and the reviewer, which helped us to improve the quality of the article. Our responses to the comments are presented in the table within the attached document titled "Response to Reviewers". All of the necessary changes, suggested by you and the reviewer are highlighted within the document titled “Revised Manuscript with Track Changes”. 

The co-authors and I hereby affirm that this manuscript has not been published previously, accepted for publication elsewhere and it is not under consideration for publication elsewhere. 

We hope that our manuscript will now be acceptable for publication in PLOS ONE following your helpful comments. 

Yours sincerely, 

Jamie Torrance (corresponding author)

---

## [Editor Report · Decision Letter 1]

2 Feb 2022

Conceptualising emotional and cognitive dysregulation amongst sports bettors; an exploratory study of ‘tilting’ in a new context

PONE-D-21-32602R1

Dear Dr. Torrance,

We’re pleased to inform you that your manuscript has been judged scientifically suitable for publication and will be formally accepted for publication once it meets all outstanding technical requirements.

Kind regards,

Rabiu Muazu Musa, PhD

Academic Editor

PLOS ONE
---

## [Editor Report · Acceptance letter]

7 Feb 2022

PONE-D-21-32602R1 

Conceptualising emotional and cognitive dysregulation amongst sports bettors; an exploratory study of ‘tilting’ in a new context 

Dear Dr. Torrance:

I'm pleased to inform you that your manuscript has been deemed suitable for publication in PLOS ONE. Congratulations! Your manuscript is now with our production department. 

Kind regards, 

on behalf of

Dr. Rabiu Muazu Musa 

Academic Editor

PLOS ONE